# *Male sterile 305* Mutation Leads the Misregulation of Anther Cuticle Formation by Disrupting Lipid Metabolism in Maize

**DOI:** 10.3390/ijms21072500

**Published:** 2020-04-03

**Authors:** Haichun Shi, Yang Yu, Ronghuan Gu, Chenxi Feng, Yu Fu, Xuejie Yu, Jichao Yuan, Qun Sun, Yongpei Ke

**Affiliations:** 1College of Agronomy, Sichuan Agricultural University, Chengdu 611130, China; haichun169@163.com (H.S.); guronghuan@xdf.cn (R.G.); yuxj305@163.com (X.Y.); yuanjichao5@163.com (J.Y.); 2Sichuan Nongda Zhenghong Bio. Co., Ltd., Chengdu 610213, China; 3Key Laboratory of Bio-resources and Eco-environment, Ministry of Education; College of Life Sciences, Sichuan University, Chengdu 610064, China; yangyu0221@139.com (Y.Y.); fcx787883757@126.com (C.F.); edible-mushroom@outloo.com (Y.F.); qunsun@139.com (Q.S.)

**Keywords:** maize, *ms305*, cuticle, transcriptomic, lipidomic, UHPLC-MS

## Abstract

The anther cuticle, which is mainly composed of lipid polymers, functions as physical barriers to protect genetic material intact; however, the mechanism of lipid biosynthesis in maize (*Zea mays. L.*) anther remains unclear. Herein, we report a male sterile mutant, *male sterile 305* (*ms305*), in maize. It was shown that the mutant displayed a defective anther tapetum development and premature microspore degradation. Three pathways that are associated with the development of male sterile, including phenylpropanoid biosynthesis, biosynthesis of secondary metabolites, as well as cutin, suberine, and wax biosynthesis, were identified by transcriptome analysis. Gas chromatography-mass spectrometry disclosed that the content of cutin in *ms305* anther was significantly lower than that of fertile siblings during the abortion stage, so did the total fatty acids, which indicated that *ms305* mutation might lead to blocked synthesis of cutin and fatty acids in anther. Lipidome analysis uncovered that the content of phosphatidylcholine, phosphatidylserine, diacylglycerol, monogalactosyldiacylglycerol, and digalactosyldiacylglycerol in *ms305* anther was significantly lower when compared with its fertile siblings, which suggested that *ms305* mutation disrupted lipid synthesis. In conclusion, our findings indicated that *ms305* might affect anther cuticle and microspore development by regulating the temporal progression of the lipidome in maize.

## 1. Introduction

Maize is a monoecious plant with separate male and female flowers growing on the same plant. Detasseling of the female inbred parent is widely employed by the seed industry to produce hybrid seed [1]. Although mechanical detasseling is effective in maize hybrid seed production, it is time-consuming and labour-intensive [2]. In addition, damage to the top leaves during detasseling reduces the hybrid seed yield [3]. Male sterility is the most efficient way to ensure cross-pollination [4,5]. Nuclear genes, mainly those that affect tapetum or microspore development in flowering plants, control genetic male sterility (GMS), which is stable in different germplasms and growth environments [6]. In maize, anther and pollen development involves complex processes with the expression of about 24,000–32,000 genes in a span of nearly 30 days [7,8]. However, only a handful of these genes have been cytologically characterized, and even fewer male sterile genes have been isolated. Moreover, over 50 GMS mutants in maize have been identified and characterized with the majority being recessive, such as *ZmMs7* [9], *Ms8* [10], *Ms9* [11], *Ms22/Msca1* [12], *Ms23* [13], *Ms26* [14], *Ms30* [15], *Ms32* [16], *ZmMs33* [17], *Ms45* [18], *APV1* [19], *IPE1* [20], *MAC1* [21], and *OCL4* [22]. The cloning and functional characterization of some male sterile genes have significantly contributed to our understanding of the molecular mechanisms of anther and pollen development in maize, which might provide useful genetic resources for hybrid maize seed production. Nevertheless, when compared with the deep understanding of the molecular mechanisms and gene networks related to anther and pollen development in the model species **Arabidopsis** and rice, comparatively little is known regarding the equivalent processes in maize [23,24].

Lipids, categorized into eight groups (fatty acyls, glycerolipids, gly-cerophospholipids, sphingolipids, sterol lipids, prenollipids, saccharolipids, and polyketides) [25], play an important role in the development of maize anthers [26,27]. The anther cuticle is a derivative of fatty acids, which is synthesized in the endoplasmic reticulum of tapetum and distributed in the epidermis of anther to protect it from hydration [26]. Sporopollenin, a biopolymer that consists of lipid monomers, is a major component of pollen exine [28]. The main lipid precursors of sporopollenin include linear fatty acids and monomers of oxygen-containing aromatic compounds, such as coumarin (C9) and ferulic acid (C10), all of which are synthesized in the tapetum cells [26]. In recent years, many genes that are involved in fatty acid metabolism and mediating anther cuticle and sporopollenin formation, whose mutation often leads to male sterile in **Arabidopsis* thaliana*, rice and maize have been reported [19,29,30]. In addition, phospholipase a (releases lysophospholipids) and lipoxygenase (catalyses fatty acid oxidation) were involved in regulating pollen germination in olive (*Olea europaea L.*) [31]. Glycosylphosphatidylinositol was reported to regulate pollen germination and tube growth in **Arabidopsis** [32]. Membrane-bound glycerol-3-phosphate acyltransferase that mediates the glycerolipid biosynthesis was regarded as being essential for tapetum differentiation and pollen development in **Arabidopsis** [33]. In spite, lipid metabolism pathway and its associated enzymes are important for the fertility of plant male gametes, the mechanism underlying the lipid metabolism regulating anther fertility in maize remains unclear.

A maize male sterile mutant of K305ms was obtained from inbred line K305 that was induced by radiation. When compared to the wild-type K305 plants, this mutant did not show a significant difference in major agronomic traits and their combining abilities, as well as the heterosis of the yield [34]. Genetic analysis indicated that a single recessive gene of *ms305* controlled the fertility traits of K305ms. Moreover, genetic mapping showed that the responsible gene was located between two simple sequence repeat markers on chromosome 2L in a region of 10.3 cM [35,36]. In this study, maize male sterile *ms305* (K305ms) and their sibling male-fertile (K305F) plants were investigated by cytological, transcriptomic, and lipidomic analysis. Through cytological observation, the abortion period and characteristics of *ms305* anthers were determined. Subsequently, RNA-seq experiment was conducted to profile the differential transcription expression between *ms305* anther and its fertile siblings at different developmental stages. Finally, the lipidomic alterations between *ms305* and fertile sibling anthers were investigated while using UHPLC-MS experiment. Our findings will enhance the mechanistic understanding of lipid biosynthesis and cuticle development in maize anther.

## 2. Results

### 2.1. Characterization of the Male Sterile Mutant ms305 Anthers

K305 male sterility plant (K305ms, ms/ms) and its sibling fertile plants (K305F, ms/Ms) were obtained from the inbred line K305 by 60Co-γ irradiation. Genetic analysis from our previous investigation indicated that a single recessive genic gene of *ms305* controlled the male sterility of K305ms [35]. When compared with the normal K305F anthers, we found that the anthers of *ms305* failed in extruding from the spikelet, and no pollen grain was produced inside anthers, leaving an empty shell (Figure 1). These results evidenced that the *ms305* anther was completely sterile.

The tissue transverse section observation found no difference in the anatomical structure or meiotic events between *ms305* and K305F anther before the dyad stage, both *ms305* and fertile sibling (K305F) anther had four somatic layers, and meiocytes were able to undergo normal meiosis (Figure 2A,B,E and F). After meiosis, tetrad cells separated into separate microspores (MP); the structure of anther wall showed a significant difference between K305F and *ms305* (Figure 2C,G). The tapetum cells of K305F anther became thinner, while the anther of *ms305* showed obvious abnormalities, including the collapse of anther wall, microspore degradation, and failure of tapetum cell degradation (Figure 2G). During the mature pollen grain stage (Figure 2D), the MP size of K305F gradually enlarged with the development and became spherical. Meanwhile, the anther wall became thinner and tapetum gradually disappeared due to programmed cell death (PCD). While MP and tapetum of *ms305* anther had completely degraded, other somatic layers of anther remained unchanged (Figure 2H). These indicated that the abnormal development of the tapetum and other anther somatic layers of *ms305* maize led to the failure of pollen formation.

Laser confocal observation showed that there was no complete nuclear structure in the anther wall of *ms305* during microspore and mature pollen cells (Figure 3A–D). Scanning electron microscopy manifested that the epidermis of *ms305* anther was smooth and lacking reticulate cuticle structure (Figure 3E–H). The mentioned results above demonstrated that blocked anther wall formation was one of the main abortion characteristics of *ms305* maize.

### 2.2. Transcriptional Differential Expression Analysis of ms305 and its Fertile Sibling Anther

Three periods of pollen mother cell, dyad, and tetrad stage (designated as I, II, III, respectively) were chosen to profile the transcriptional differential expression between *ms305* (M) and its fertile sibling (F) plant, according to the cytological analysis results. Firstly, six genes were randomly selected for qPCR detection. The results showed that the results of RNA-seq were in accordance with the qPCR analysis (Appendix A), which indicated that the RNA-seq data met the requirements for further transcriptome analysis. We uncovered, respectively, 451, 108, and 1435 differential expressed genes (DEGs) in three stages (Appendix A). Among them, 34 DEGs were commonly expressed in all stages, including xyloglucan endotransglucosylase, O-methyltransferase ZRP4-like rotease inhibitor, two metacaspase family proteins, two dehydrogenases, and a peroxidase (Appendix A), which involved in cell wall formation, programmed cell death (PCD), and fatty acid metabolism, which might be associated with infertility. Notably, an aldehyde dehydrogenase gene (ALDH, GRMZM2G090245) in common DEGs showed the highest down regulation in the dyads (log FC 8.37) and tetrad (log FC 7.84) stage of *ms305* anther (Appendix A).

KEGG enrichment analysis found that three pathways, including ‘phenylpropanoid biosynthesis’, ‘biosynthesis of secondary metabolites’, and ‘cutin, suberine, and wax biosynthesis’, were commonly enriched in all three development stages of anther (Figure 4A). These pathways were associated with cell wall and tapetum metabolism, which confirms the results of cytological observation.

It is clear that wax and cutin are essential components of anther epidermis, which are vital synthetic products of fatty acid and lipid metabolism. In this study, 11 DEGs for ‘cutin, suberine, and wax biosynthesis’ and 42 DEGs for ‘lipid metabolism’ were identified (Figure 4B, Appendix A). We used the online database STRING and Cytoscape to further construct the (protein–protein interaction) PPI network among these genes, and a molecular interaction of 25 genes were identified (Figure 5C). The hub genes of higher degree (a node connects with other nodes), which play a key role in the network, were identified using Cytohubba in Cytoscape. Appendix A lists tge degree of the hub genes. The top five hub genes were a 3-oxoacyl-[acyl-carrier-protein] synthase II (KAS II, GRMZM2G124335), an ALDH gene (GRMZM2G118800), a Glucose-6-phosphate 1-dehydrogenase (G6PD, GRMZM2G031107), a lactate dehydrogenase (LDH, GRMZM2G173192), and a mannitol dehydrogenase (MDH, GRMZM2G118610).

### 2.3. Aliphatic Alteration of ms305 Anther

Wax and cutin relate to tapetum metabolism and they participate in the formation of pollen wall. The content of wax and cutin in anthers was detected by GC-MS, according to results of transcriptome analysis. When compared with fertile siblings, the content of cutin in *ms305* anther significantly decreased in the pollen mother cell, and tetrad stages (*p* < 0.05), while no significant change was observed in wax content (*p* > 0.05; Figure 5A). Further analysis of cutin monomer showed a significant decrease in *ms305* anther (*p* < 0.05; Figure 5B). These results suggested that wax and cutin metabolism might lead to the abortion of *ms305* maize.

GC-MS determined the change of soluble fatty acids between *ms305* and its fertile sibling anther. From pollen mother cell to microspore stage, the total fatty acid content in fertile sibling increased, while that of sterile line *ms305* gradually decreased (Figure 5C). Especially during the microspore stage, the total amount of fatty acids in fertile sibling was 1.55 mg/g, while that of *ms305* was only 0.95 mg/g (*p* < 0.01). A total of 15 different fatty acid molecules were found in this stage. Linoleic acid (C18:2), one of the most important fatty acids in maize anthers, was down-regulated nearly 19-fold in *ms305* anther (Figure 5D). These results indicated that the fatty acid metabolism pathway is involved in the regulation of *ms305* anther abortion.

### 2.4. Lipids Alteration of ms305 Anther

We further investigated the difference of lipid metabolism in the microspore stage between *ms305* and fertile sibling anther by UHPLC-MS. A total of 25 lipid classes were identified in the anthers of the two materials, as shown in the Appendix A. Glycerolipids, glycerophospholipids, and sphingolipids were the main lipid derivatives of maize anthers; however, their composition structures were significantly different between the fertile and sterile plant (Appendix A). When compared with fertile sibling, the glycerolphospholipids of *ms305* anther, such as lysophosphatidic acids (LPA), phosphatidylcholine (PC), lysoPC (LPC) phosphatidylserines (PS), and phosphatidylinositols (PI), showed a significant down-regulation (*p* < 0.05; Figure 6). Further analysis of all lipid species revealed that half of the 127 differentially identified lipid species belonged to glycerolipids of 45 triacylglycerols (TGs) and 15 digalactosyldiacylglycerols (DGs); among the 15 DGs, 13 of them were down-regulated (Figure 7; Appendix A). Additionally, totally, 16 glycerophospholipids were identified, including seven PCs, four PSs, one phosphatidic acid (PA), one phosphatidylethanolamine (PE), one phosphatidylglycerol (PG), one PI, and one LPC; among all glycerophospholipids, 14 of them were down-regulated (Figure 7). In addition, seven differentially expressed galactolipids were identified as three monogalactosyldiacylglycerols (MGDG) and four digalactosyldiacylglycerols (DGDG), all of which were down-regulated (Figure 7). The up-regulated lipid species mainly concentrated in ceramidase (Cer) and TG, including 26 Cers and 24 TG lipid molecules (Appendix A).

## 3. Discussion

There are many types of abortion in maize GMS mutants, such as anther wall layer defects [15,16], abnormal meiosis of pollen mother cells [10], abnormal anther dehiscence [37], and abnormal callose degradation [38]. In the present study, the development of four somatic layers (epidermis, chamber wall, middle layer, and tapetum) of *ms305* were delayed, and microspore abnormally degraded. No mutant with the same abortion characteristics as *ms305* has been reported in maize to the best of our knowledge. Our previous genetic mapping showed that male sterile gene of *ms305* was located between two simple sequence repeat markers on chromosome 2L in a region of 10.3 cM, bnlg469b, and bnlg1940 [35]. Among the DEGs that were identified in ‘cutin, suberine, and wax biosynthesis’ and ‘lipid metabolism’ pathway, a KAS II (GRMZM2G124335) gene was found to be located in this region (Figure 4C, Appendix A); interestingly, this gene was the hub gene of highest degree value in the PPI network analysis; thus, further map-based cloning of KAS II will be helpful. Notably, the male sterile gene *Ms33* was located in this region, which encodes a glycerol-3-phosphate acyltransferase protein that mediates anther cuticle formation and microspore development [17,39]. Sequence analysis of *ms305* genomic DNA revealed a 6-bp deletion in *Ms33* (unpublished data). However, the abortion characteristics of *Ms33*, which showed no difference in anatomical structure or meiotic events in the fertile and sterile plants before the tetrad stage, as well as the severe degradation of the tapetum in sterile plants at the early uninucleate stage [39], were completely opposite to those of *ms305,* which delay the degradation of tapetum at this stage. Further cloning of *ms305* is necessary to confirm whether *ms305* is a novel male sterile mutant or an allele of *Ms33* in maize.

Two metacaspase family proteins (GRMZM2G132238 and GRMZM2G120079), which were discovered to be family numbers of caspase homologs that are involved in PCD processes during plant development, were specially down regulated in *ms305* anther during three development stages of anther [40]. Reactive Oxygen Species (ROS) were reported to regulate PCD in plant [41], and a large down regulation of peroxidase (GRMZM2G471357) in K305ms anther suggested its potential role in PCD and anther development. We hypothesize that delayed PCD of the inner four anther layers in *ms305*, including tapetum cells, leads to the degradation of pollen grains, and this should be responsible for its complete male sterility.

Lipids play an important role in regulating plant anther development, a large number of studies have reported the role of fatty acids and their derivatives in regulating anther fertility [15,29,42]. For example, anther cuticle, including cutin and wax, can protect plant anthers from hydration [26]. In maize, *MS45*, *MS26*, *MS30*, *MS33*, *IPE1*, and *APV1* are reported to participate in the aliphatic metabolic pathways, which are required for anther cuticle development [19,26,29,39,43]. The abortion characteristics of *MS30*, *MS33*, *IPE1*, and *APV1* in maize were described, such as the swollen microspore and the decreased vacuole of *APV1* at the vacuolated stage [19], severe degradation of the tapetum in *IPE1* [20] and an obvious abortion of *MS30* first appeared after the vacuolated stage [15], these were, however, different from *ms305*. In our study, *ms305* also seriously affected the synthesis of anther cuticle and its monomers, as well as the content of soluble fatty acids in anthers. In addition, among the 11 DEGs in anther cuticle biosynthesis pathway, six belonged to the cytochrome P450 (CYP450) superfamily (Figure 4B, Appendix A), and notably, *MS26* and *APV1* belong to the CYP450 subfamily member, suggesting the important role of these six DEGs. For the two ALDH genes of GRMZM2G090245 and GRMZM2G118800 in the lipid metabolism pathway, the former was found to be down regulated in *ms305* anther during all three development stages (Appendix A), while the latter was one of the top three hub genes in the PPI network (GRMZM2G118800, Figure 4B); and, previous studies found that ALDH activity was required for male fertility in plants [44,45]. We also found six GDSL lipases from the 42 DEGs in the lipid metabolism pathway, coincidentally, *MS30* was reported to encode a GDSL lipase [15]. Taken together, these ALDH, GDSL lipases, and CYP450 superfamily genes would be involved in regulating lipid metabolism pathways as well as the cuticle metabolism in maize.

PE tends to form hexagonal II phase or other non-bilayer phases, and PC forms bilayers. Higher PC: PE ratios reduce the propensity of membranes to form non-bilayer phases. In the present study, the membrane fluidity in the pollen grains of *ms305* maize might be affected by its decreased PC: PE ratios, which was consistent with a previous report that the suppression of aminoalcohol: aminophosphotransferases catalyzed the final step of PC synthesis, thus resulting in pollen sterility in **Arabidopsis** [46]. Thus, we hypothesize that *ms305* might affect the fertility of maize pollen by participating in the regulation of PC synthesis. PS plays an important role in cell death signaling, vesicular trafficking, lipid–protein interactions, and membrane lipid metabolism [47]. Previous studies found PS synthase1 was important for microspore maturation in **Arabidopsis* thaliana*, and dynamic PS distribution in nuclei and different vesicular compartments was also important during microspore maturation [48]. The significant downregulation of PS in the *ms305* anther indicated that PS played an important role in regulating maize pollen fertility.

Galactolipids mostly occur in plastidial membranes. Studies on the glycerolipid composition of *B. napus* pollen found that almost galactolipids are distributed in the pollen coat [49]. So far, the role of galactolipids in plant pollen development remains unclear. However, the MGDG synthase inhibitor galvestine-1 was applied to pollen in growth medium, which resulted in a halving of pollen growth [50], indicating the importance of galactolipids synthesis in growing pollen tubes. We speculate that the change of galactolipid content is one of the causes of pollen abortion in *ms305* maize, while the regulation mechanism needs to be further studied. In addition, DG, as an important signal molecule in plants, is the substrate of MGDG and DGDG synthesis, while PC is the precursor of DG synthesis [49]. These lipid molecules are down-regulated in sterile plants, which suggests that the lipid metabolism blocked by ms305 mutation is one of the main factors that contributes to male sterility in maize.

## 4. Materials and Methods

### 4.1. Plant Materials

The sibling population of K305 male sterility, including K305 male sterility plants (K305ms, ms/ms) and K305 fertile plants (K305F, ms/Ms), were both obtained from inbred line K305 that was exposed to ^60^Co-γ irradiation [35]. A segregation sibling population of 230 plants, which was constructed by crossing homozygous male sterile plants and heterozygous male fertile plants, was planted in Shuangliu, Sichuan Province, China (30_34°N, 103_56°E) in 2014.

The samples were taken during anther developing stage. The sampling time is from 8 am to 10 am. The well-grown plants were randomly selected with gently pinching the spikes by hand, and a sterile blade was used to quickly break down the spikes in the soft feeling place. The tassels that meet the requirements were removed and immediately placed into the corresponding numbered centrifuge tubes.

The tassels that were used for cytological observation were put into a centrifuge tube containing Carnoy I fixed solution. After 8 h fixed at the room temperature, it was stored in the 70% ethanol solution and placed in the 4 °C refrigerator for subsequent experiments. The anther collected for transcriptome analysis was placed in a centrifuge tube and soaked in 10 times larger volume of RNAwait protection solution (Solarbio, Beijing, China) than itself. Afterwards, the tube was preserved in an ice box temporarily and then transferred to 4 °C refrigerator once brought back to laboratory for storage overnight then −20 °C for cryopreservation. After the fertility identification of the plant, the material was scrutinized. The anther was stripped from the spikelet with tweezers and anatomic needles, and the length was measured.

### 4.2. Phenotypic Analysis of ms305

Fresh anthers from both wild type and the mutant at different stages were immersed in FAA solution (50% ethanol, 5% glacial acetic acid, 5% formalin) for 24 h at room temperature for fixation for scanning electron microscope (SEM) analysis. The samples were then dehydrated in a serial of ethanol gradients (50–100%). After critical-point drying, the anthers were coated with palladium gold and then observed using a scanning electron microscope (HITACHI S-3400N, Tokyo, Japan).

For cytological observation, the anthers were pricked and fixed in FAA solution overnight. The samples were dehydrated using a serial of ethanol (50–100%) and embedded in spurr resin. Semi-thin sections were obtained while using a Leica UE (Leica Microsystems, Berlin, Germany), stained with 0.05% toluidine blue and observed with an Olympus BX-53 microscope (Tokyo, Japan). For laser scanning confocal microscopy analysis, fresh anthers were vacuum infiltrated and prefixed in 3.5% glutaraldehyde (0.1 M phosphate buffer, pH 7.4), followed by rinsing with 0.1 M phosphate buffer. The samples were transferred into 1% osmium tetraoxide and rinsed with 0.1 M phosphate buffer. After fixation, the samples were dehydrated using an ethanol series from 50% to 100% and then embedded in spurr resin. Ultra-thin sections were collected with a Leica EM-UC6 (Leica Microsystems, Berlin, Germany).

### 4.3. Aliphatic Components Analysis

The anthers were considered as cylinders to calculate the surface area. The anther surface area was plotted against the corresponding fresh weight. Cuticle wax, cutin and total soluble fatty acid extraction and Gas chromatography-mass spectrometry (GC-MS) analysis were performed, as described by Shi, et al. [51].

### 4.4. RNA Extraction and qRT-PCR

The total RNA was isolated from different stages of anthers using the RNeasy Plant mini Kit (QIAGEN, Hilden, Germany) as described by the manufacturer. The developmental stage of anthers was determined based on the semi-section morphology. One microgram of total RNA was used to synthesis cDNA using RevertAid First Strand cDNA Synthesis Kit (THERMO, UK). qRT-PCR was performed on the Roche LightCycle480 system (Roche Applied Science, Switzerland) with SYBR Green Premix (TAKARA Code RR820A, TAKARA, Shiga, Japan). All of the PCR reactions were conducted using 40 cycles at 98 °C for 10 s, 60 °C for 10 s, and 72 °C for 10 s, in a 20 µL reaction mixture containing 10 pmol of each primer and 2 µL of cDNA as template. All of the reactions were performed in triplicate, and ZmActin1 was used as the internal control for normalization. Appendix A lists the primers used for qRT-RCR.

### 4.5. Transcriptome Analysis

The total RNA was isolated from different stages of anthers while using the RNeasy Plant mini Kit (QIAGEN, Hilden, Germany), as described by the manufacturer. The developmental stage of anthers was determined based on the semi-section morphology. Eighteen cDNA libraries were constructed in accordance with standard Illumina TruSeq instructions and sequenced while using an Illumina Genome Analyzer (Hiseq 2500; Illumina, San Diego, CA, USA). High-quality clean reads were obtained by filtering the raw reads, and then maize reference genome (AGPv3; MaizeSequence.org) and TopHat2 were used for mapping. RPKM was used to calculate the gene expression level. Differentially expressed genes (DEGs) were detected by edgeR package61, which were defined according to the criteria, as follows: more than a two-fold change and FDR less than 0.05. Gene ontology (GO) enrichment and KEGG pathway enrichment analyses were performed using the Bioconductor tool. The Search Tool for the Retrieval of Interacting Genes (STRING) database (https://string-db.org/cgi/input.pl) was employed in order to construct the protein–protein interaction (PPI) network. The network was visualized in Cytoscape and the hub genes in thr network were identified by APP Cytohubba [52].

### 4.6. Lipidome Analysis

After the samples slowly thawed at 4 °C, 100mg samples were taken, homogenized with 200 µL water, and then adding 240 µL pre-cooled methanol (34,860, Sigma-Aldrich) and 800 µL MTBE (306,975, Sigma-Aldrich), vortex mixing and ultrasonic in low-temperature water bath. The upper organic phase was taken, and blow-dried by nitrogen, and 200 µL 90% isopropanol/acetonitrile solution was added for the mass spectrometry analysis, centrifuged at 14,000 g 10 °C for 15 min., and the supernatant was then taken for analysis.

The samples were processed using UHPLC Nexera LC-30A UHPLC system (Shimadzu, Kyoto, Japan) on a C8 reverse phase column, an exactive mass spectrometer UHPLC Nexera LC-30A UHPLC system was employed to separate the samples, and the ACQUITY UPLC CSH C18 column was used. Mass spectrometry was performed by Q Exactive plus mass spectrometer (Thermo Scientific™, CA, USA). Positive and negative ion modes of electrospray ionization (ESI) were used. The ESI source conditions are as follows. Positive: Heater Temp 300 °C, Sheath Gas Flow rate 45 arb, Aux Gas Flow Rate15 arb, Sweep Gas Flow Rate 1arb, spray voltage 3.0KV, Capillary Temp 350 °C, S-Lens RF Level 50%. MS1 scan ranges: 200–1800. Negative: Heater Temp 300 °C, Sheath Gas Flow rate 45arb, Aux Gas Flow Rate 15arb, Sweep Gas Flow Rate 1arb, spray voltage 2.5KV, Capillary Temp 350 °C, S-Lens RF Level 60%. MS1 scan ranges: 250–1800. The mass charge ratio of lipid molecules to lipid fragments was collected, as follows: 10 fragment profiles (MS2 scan, HCD) were collected after each full scan. MS1 has a resolution of 70,000 at M/Z 200 and MS1 was 17,500 at M/Z 200.

LipidSearch software version 4.1 (Thermo Scientific™, CA, USA) was used for peak identification, lipid identification (secondary identification), peak extraction, peak alignment, and quantitative processing. The software simca-p 14.1 (Umetrics, Umea, Sweden) was applied for pattern recognition, and the data were pretreated with pareto-scaling in order to conduct multidimensional statistical analysis. Qualitative and quantitative analysis of the obtained data from positive and negative ion modes was conducted using LipidSearch software version 4.1. The raw data for lipidome analysis were listed in Appendix A.

### 4.7. Statistical Analysis

The data of the anther cuticle, fatty acids, and lipids analysis were obtained, and then analysis of variance (ANOVA) was conducted by using the Statistical Package for Social Science (SPSS; SPSS Inc., Chicago, IL, USA) version 22.0. Statistical significance was judged at a threshold of *p* < 0.05.

## 5. Conclusions

In this study, we integrated cytological, transcriptomics, and metabolomics analysis to revel the effects of male sterile mutation *ms305* on maize anther development. The present study should be the first one to apply lipidomics analysis to reveal the mechanisms of male anther fertility in maize, which disclosed that the expression of a series of lipid molecules was inhibited in *ms305* to a varied degree, which suggested that *ms305* might participate in the regulation of anther cuticle and microspore formation by affecting the lipid metabolism at a large scale. The abortion characteristics of *ms305* were different from those of the GMS mutants previously reported, and whether *ms305* is a novel male sterile mutant or allelic to *Ms33* still needs to be further confirmed. This information will be helpful for the efficient utilization of male sterile *ms305* for maize breeding in the future.

## Figures and Tables

**Figure 1 ijms-21-02500-f001:**
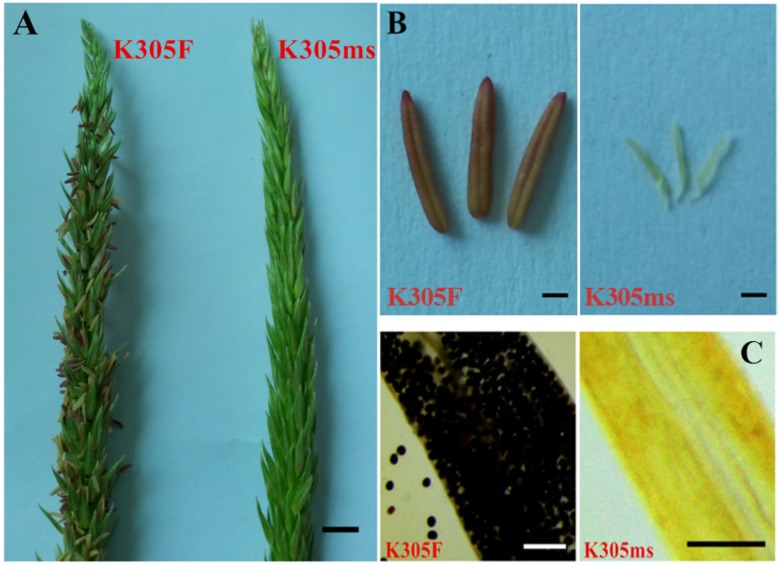
Phenotypic comparison of *ms305* and fertile sibling. (**A**) The inflorescence of *ms305* and its fertile sibling plant, scale bars = 10 mm; (**B**) The anther of *ms305* and its fertile sibling plant, scale bars = 1 mm; (**C**) Viable pollen grains from fertile sibling and nonviable pollen grains from *ms305* after I2-KI staining, scale bars = 1 mm. Fertile sibling, K305F; *ms305*, K305ms.

**Figure 2 ijms-21-02500-f002:**
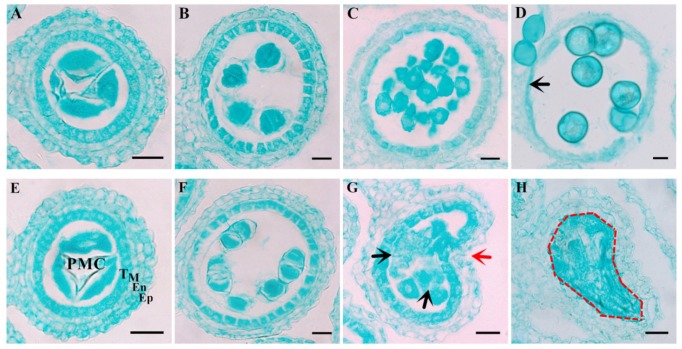
Cytological comparison of anther development in *ms305* and fertile sibling at different stages. (**A**–**D**) Anther transverse sections of different developmental stages in fertile sibling; (**E**–**H**) Anther transverse sections of different developmental stages in *ms305*. Notes: PMC, pollen mother cell; T, tapetum; M, middle layer; En, endothecium; Ep, epidermis. Red arrow, collapsed anther wall; Black arrow, degraded microspore. Dotted box, degraded microspore and tapetum cell. Scale bars = 50 μm.

**Figure 3 ijms-21-02500-f003:**
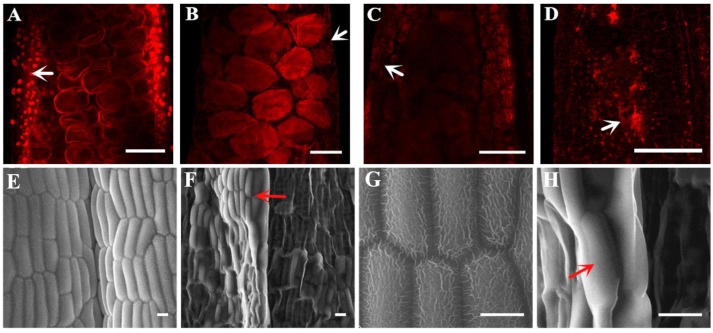
Defective development of the *ms305* anther wall. Laser scanning confocal microscopy analysis of fertile sibling (**A**,**B**) and *ms305* anther (**C**,**D**), (**A**) Microspore stage of fertile sibling; (**B**) Mature pollen stage of fertile sibling; (**C**) Degraded anther of *ms305*, no cell nucleuses were observed; (**D**) Aborted anther of *ms305*. Scale bars = 100 μm. SEM analysis of the anther surface of fertile sibling (**E**) and *ms305* (**F**), ×400; (**G**) Epidermal cell surface of fertile sibling anther with reticulate pattern, ×1500; (**D**) Epidermal cell surface of *ms305* was smooth without reticulate pattern, ×1500. Scale bars = 20μm.

**Figure 4 ijms-21-02500-f004:**
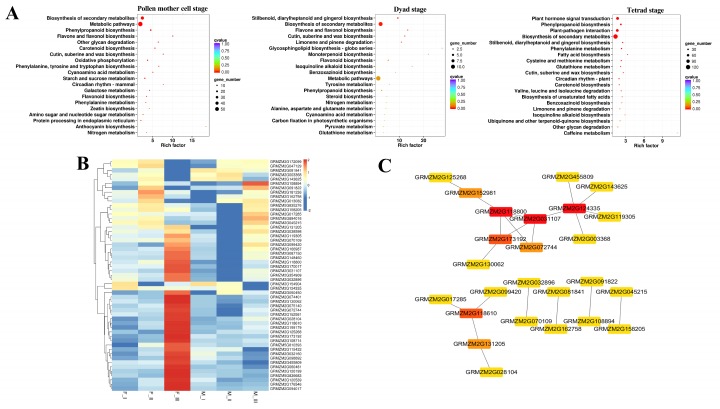
(**A**) KEGG enrichment analysis in *ms305* and fertile sibling anther at different development stages. (**B**) Heat maps of differentially expressed genes in ‘cutin, suberine, and wax biosynthesis’ and ‘lipid metabolism’ pathway; I, II, and III represent PMC, dyad, and tetrad stage (III), respectively; F, fertile sibling; M, *ms305*. (**C**) Molecular interaction network conducted by using Cytoscape, the darkness for the node degree of genes.

**Figure 5 ijms-21-02500-f005:**
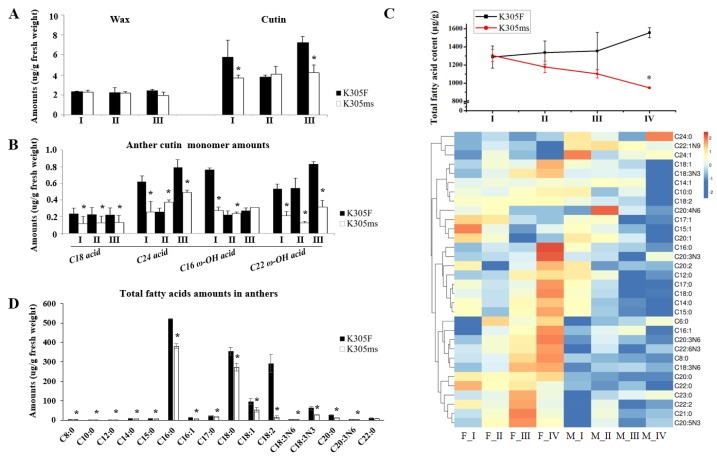
Aliphatic alteration of *ms305* anther. (**A**) Cutin and wax amounts of *ms305* (K305ms) and fertile sibling (K305F) in pollen mother cell (I), dyads (II), and tetrad stages (III), respectively. (**B**) The major differential cutin monomers between *ms305* and fertile sibling anther. (**C**) Total fatty acids of *ms305* and fertile sibling anthers in different development stages. (**D**) Alteration of *ms305* fatty acids amounts in microspore stage. I, pollen mother cell; (II), dyads; (III), tetrad; (IV) microspore. Error bars indicate SD (*n* = 5); **p* < 0.05.

**Figure 6 ijms-21-02500-f006:**
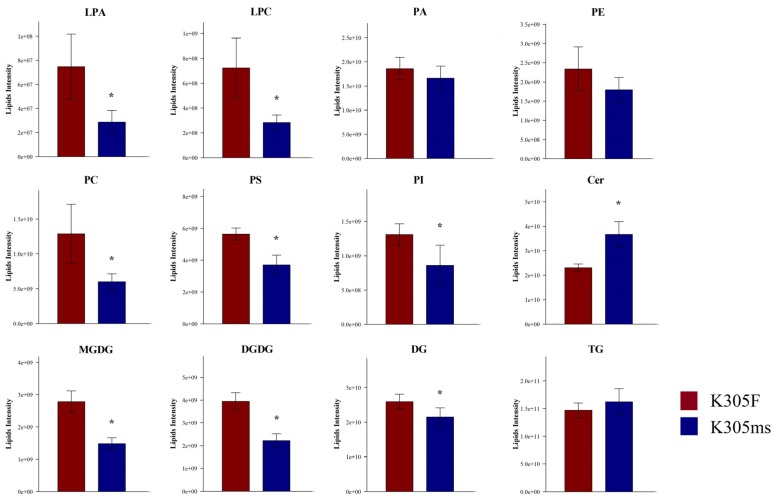
UHPLC-MS analysis of total lipids in *ms305* (K305ms) and fertile sibling (K305F). PA, phosphatidic acid; LPA, lysoPA; PC, phosphatidylcholine; LPC, lysoPC; PE, phosphatidylethanolamine; PS, phosphatidylserines; PI, phosphatidylinositols; MGDG, monogalactosyldiacylglycerols; DGDG, digalactosyldiacylglycerols; DG, digalactosyldiacylglycerols; TG, triacylglycerols; Cer, ceramidase. Error bars indicate SD (*n* = 8); * *p* < 0.05.

**Figure 7 ijms-21-02500-f007:**
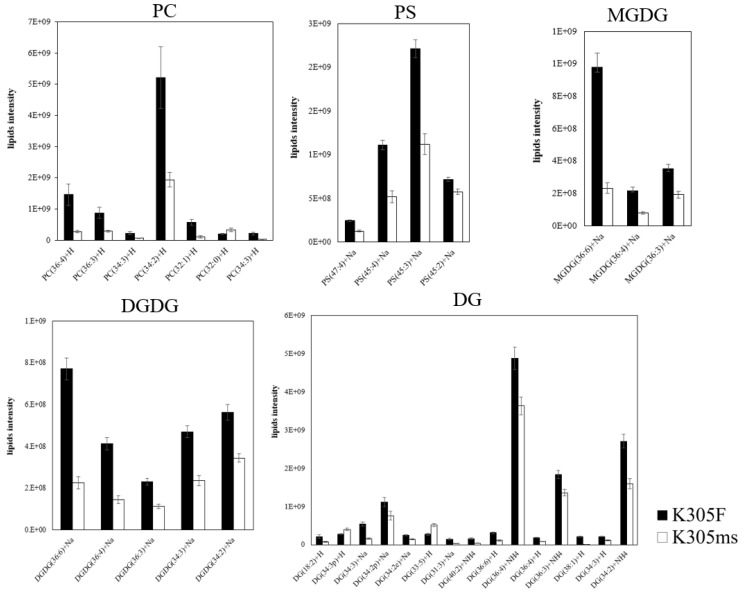
Differentially expressed lipid molecules between *ms305* (K305ms) and fertile sibling (K305F) anthers. PC, phosphatidylcholine; PS, phosphatidylserines; MGDG, monogalactosyldiacylglycerols; DGDG, digalactosyldiacylglycerols. Error bars indicate SD (*n* = 8).

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
