# Peer review of "Male sterile 305 Mutation Leads the Misregulation of Anther Cuticle Formation by Disrupting Lipid Metabolism in Maize"

_ijms, 2020, doi:10.3390/ijms21072500_

Round 1

Reviewer 1 Report

In this manuscript, authors characterized Maize male-sterile mutant ms305 by phenotypic, transcriptomic and lipidomic analysis. The data presented here seem solid, however, some minor corrections are required.

1) What kind of statistics was used in this study? Authors should be described it.

2) I assume ms305 mutant is a single, recessive mutation, because authors mentioned that both K305F and ms/Ms plants are fertile in the section 4.1. This should be also described in the Results section.

3) Manuscript should be checked thoroughly to correct errors. Below are some (not all) examples:

Page 1, lines 22 and 26.

…stage (p < 0.05), …siblings (p < 0.05)

Description of p value is not necessary in the abstract.

Page 6, lines 173-174.

…25 lipid classes and were… -> delete “and”?

Page 7, line 173, page 8, line 214, and page 8, line 250.

We hypothesis -> We hypothesize

Author Response

Response to Reviewer 1 Comments

Dear Reviewer:

Thank you for your comments on our manuscript “Male sterile 305 mutation leads the misregulation of anther cuticle formation by disrupting lipid metabolism in maize” (ijms-746503). We have finished a thorough revising work according to the comments (modifications are highlighted in yellow in the revision), and an item by item response is as follows:

Point 1: What kind of statistics was used in this study? Authors should be described it.

Response 1:

The data of the anther cuticle, fatty acids and lipids analysis were subjected to an analysis of variance (ANOVA) y using the Statistical Package for Social Science (SPSS; SPSS Inc., Chicago, IL, USA) version 22.0. The statistical significance was judged at a threshold of P < 0.05. We have added this information in Page 11, lines 375-378.

Point 2: I assume ms305 mutant is a single, recessive mutation, because authors mentioned that both K305F and ms/Ms plants are fertile in the section 4.1. This should be also described in the Results section.

Response 2:

Thanks for your kind reminder. K305 male sterility plant (K305ms, ms/ms) and its sibling fertile plants (K305F, ms/Ms) were obtained from inbred line K305 by 60Co-γ irradiation. Genetic analysis from our previous investigation indicated that the male sterility of K305ms was controlled by a single recessive genic gene of ms305. We have added this information in Page 2, lines 85-88.

Point 3: Manuscript should be checked thoroughly to correct errors. Below are some (not all) examples:

Page 1, lines 22 and 26.

…stage (p < 0.05), …siblings (p < 0.05)

Description of p value is not necessary in the abstract.

Page 6, lines 173-174.

…25 lipid classes and were… -> delete “and”?

Page 7, line 173, page 8, line 214, and page 8, line 250.

We hypothesis -> We hypothesize

Response 3:

We have examined the manuscript and corrected some errors, as follows:

Page 1, lines 22 and 26.

We have deleted the description of p value in the abstract.

Page 6, lines 188-189.

a total of 25 lipid classes and were identified in the anthers of the two materials. “and” was deleted.

Page 7, line 235, and page 8, line 268.

 “We hypothesis” was corrected as “We hypothesize”.

Reviewer 2 Report

The manuscript by Shi and co-workers attempts to characterize a male sterile line (ms305) in maize. The study aims to characterize the underlying biochemistry of tapetum development in ms305 genotype. The study also reveals the structural basis of tapetum walls and composition of lipids, which forms the bulk of anther cuticle. The report utilizes a set of tools including microscopy, RNA-seq and lipid profiling strategies. The strengths of this study trust on the ability to demonstrate the differences in lipid compositions between ms305 and fertile sibs. The results and discussion section is appropriately written.

The locus conferring male sterility was mapped to a broader region spanning 10.3cM on chromosome 2L. However, it is disappointing that the genetic locus responsible for male sterility is not reported. The assertions (line 344) on revealing molecular mechanism appear an overstatement because the conclusions are based on morphological and lipid profiles. Though the male sterility phenotype was identified in the inbred line exposed to gamma radiated, the nature of mutation remains unknown.  It will be interesting to clone and characterize the gene controlling fertility.

The RNA-seq analyses may reveal misregulated genes (Line 141-145) represented in various lipid, wax, and secondary metabolite pathways, and these may also support the measurements on diverse lipids/metabolite levels, however, offers very little understandings on the consequences in biological perspective. It would be interesting to report any differentially expressed genes located within the 10.3cM interval.

Minor comments: 1. The legend for Fig.6 should provide full form of abbreviations.

2. The axis labels on Fig. S2 needs to be placed appropriately to make it readable

Author Response

Response to Reviewer 2 Comments

Dear Reviewer:

Thank you for your comments on our manuscript “Male sterile 305 mutation leads the misregulation of anther cuticle formation by disrupting lipid metabolism in maize” (ijms-746503). We have finished a thorough revising work according to the comments (modifications are highlighted in yellow in the revision), and an item by item response is as follows:

Point 1: The locus conferring male sterility was mapped to a broader region spanning 10.3cM on chromosome 2L. However, it is disappointing that the genetic locus responsible for male sterility is not reported. The assertions (line 344) on revealing molecular mechanism appear an overstatement because the conclusions are based on morphological and lipid profiles. Though the male sterility phenotype was identified in the inbred line exposed to gamma radiated, the nature of mutation remains unknown.  It will be interesting to clone and characterize the gene controlling fertility.

Response 1:

In this study, we integrated cytological, transcriptomics, and metabolomics analysis to revel the effects of male sterile mutation ms305 on maize anther development, and disclosed that the expression of a series of lipid molecules was inhibited in ms305 to varied degree, suggesting that ms305 might participate in the regulation of anther cuticle and microspore formation by affecting the lipid metabolism at a large scale. Cloning and characterize of key genes in this pathway is necessary, and this work is being studied in the next study, because the use of such genes plays an important role in maize breeding.

The assertions of revealing molecular mechanisms by lipidomics analysis is exaggerated, we have revised this conclusion, which might be more appropriate. See details in Page 11, line 380-385.

Point 2: The RNA-seq analyses may reveal misregulated genes (Line 141-145) represented in various lipid, wax, and secondary metabolite pathways, and these may also support the measurements on diverse lipids/metabolite levels, however, offers very little understandings on the consequences in biological perspective. It would be interesting to report any differentially expressed genes located within the 10.3cM interval.

Response 2:

Thanks for your advice. It has greatly improved our work. According to the result of RNA-seq analyses, 11 DEGs for ‘cutin, suberine, and wax biosynthesis’ and 42 DEGs for ‘lipid metabolism’ were identified (Figure 4B, Table S3). We used the online database STRING and Cytoscape to further construct the (protein–protein interaction) PPI network among these genes, and a molecular interaction of 25 genes were identified (Figure 5C). The hub genes of higher degree (a node connects with other nodes), which play a key role in the network, were identified by using Cytohubba in Cytoscape.

Among the DEGs identified in ‘cutin, suberine, and wax biosynthesis’ and ‘lipid metabolism’ pathway, a KAS II (GRMZM2G124335) gene was found to locate in the 10.3cM interval (Figure 4C, Table S3); interestingly, this gene was the hub gene of highest degree value in the PPI network analysis, thus, further map-based cloning of KAS II will be helpful.

In addition, among the 11 DEGs in anther cuticle biosynthesis pathway, six were belong to cytochrome P450 (CYP450) superfamily (Figure 4B, Table S3), and notably, maize GMS mutants MS26 and APV1 to CYP450 subfamily member, suggesting the important role of these six DEGs. For the two ALDH genes of GRMZM2G090245 and GRMZM2G118800 in lipid metabolism pathway, the former was found to be down regulated in ms305 anther during all three development stages (Table S2), while the latter was one of the top three hub genes in PPI network (GRMZM2G118800, Figure 4B); and previous studies found that ALDH activity was required for male fertility in plants. We also found six GDSL lipases from the 42 DEGs in lipid metabolism pathway, coincidentally, MS30 was reported to encode a GDSL lipase. Taken together, these ALDH, GDSL lipases and CYP450 superfamily genes would be involved in regulating lipid metabolism pathways as well as the cuticle metabolism in maize.

This information have been add in the revised manuscript, see details in Page 5, lines 152-162, Page 7, lines 218-221, Page 8, lines 243-258.

Point 3: Minor comments: 1. The legend for Fig.6 should provide full form of abbreviations.

  1. The axis labels on Fig. S2 needs to be placed appropriately to make it readable

Response 3: Some minor corrections:

  1. Some missed abbreviations in the legend for Fig.6 and Fig. 7 were added. See details in Page 7 and 8.
  2. The axis labels on Fig. S2 has been adjusted to its proper position. See details in Page 13.

Reviewer 3 Report

The authors should be required for consideration of the following points to revise the manuscript.

General comments:

  • .Summarizing and presenting omics data could be challenging task. I would suggest mapping both transcriptome and lipidome data to the biosynthesis pathways by using Cytoscape or a similar software. Adding such figure will provide an informative overview of the pathways affected by the mutation and could be even helpful to identify the causal gene(s).
  • P7, R225: “…For instance, aldehyde dehydrogenase showed a remarkably down regulation in dyads (70-fold) and tetrad (61-fold) stage of ms305 anther…” This data is not presented in the manuscript. Table S2 shows only the DEGs that are common for the three studied anther development stages. The authors should provide additional supplemental tables with the identified DEGs in each developmental stage. In addition, both RNA-seq and lipidomics data should be submitted to public databases and accession numbers cited in the manuscript.
  • P6, R205: “…However, the abortion characteristics of ms305 were different from Ms33,…” Please, describe the differences in more details. The observed phenotype of ms305 should also be compared to other known MS mutants with affected lipids biosynthesis and the differences highlighted and discussed to further support the novelty of ms305 mutant.
  • English style and grammar needs further improvement. See some more specific comments below:

-P2, R66: regard -> regarded

-P2, R67: “In spite lipid metabolism and the enzymes involved in those 67 pathways are important for the fertility of plant male gametes, the mechanism underlying the lipid metabolism regulating anther fertility in maize remains unclear.”- Not clear, consider revising this sentence

-P2,R73: delete “genic”

-P5, R144: “which confirming”->”which confirms”

-P6, R199: “…So far, no literature reported on maize abortion characteristics is consistent with that of male sterile mutants ms305.”-Consider revising to clarify

-P6, R205 “…which showed a severe degradation of the tapetum at the early uninucleate stage [32]…”-Check the reference [32}. Seems to be incorrect in the context.

-P7, R237: hypothesis-> hypothesize

-P8, R265: “…sterile blade was used to quickly break down the spikes in the soft-smelling place.” Please clarify what is soft-smelling place.

-P8, R268: was->were

Author Response

Response to Reviewer 3 Comments

Dear Reviewer:

Thank you for your comments on our manuscript “Male sterile 305 mutation leads the misregulation of anther cuticle formation by disrupting lipid metabolism in maize” (ijms-746503). We have finished a thorough revising work according to the comments (modifications are highlighted in yellow in the revision), and an item by item response is as follows:

Point 1: Summarizing and presenting omics data could be challenging task. I would suggest mapping both transcriptome and lipidome data to the biosynthesis pathways by using Cytoscape or a similar software. Adding such figure will provide an informative overview of the pathways affected by the mutation and could be even helpful to identify the causal gene(s).

Response 1:  Thanks for your suggesting.

We used the online database STRING and Cytoscape to construct the (protein–protein interaction) PPI network among the DEGs from cuticle and lipid metabolism pathways, and a molecular interaction of 25 genes were identified (Figure 5C). The hub genes of higher degree (a node connects with other nodes), which play a key role in the network, were identified by using Cytohubba in Cytoscape. A KAS II (GRMZM2G124335) gene was found to locate in the 10.3 cM region of ms305 (Figure 4C, Table S3); interestingly, this gene was the hub gene of highest degree value in the PPI network analysis. An ALDH gene was identified as one of the top three hub genes in PPI network (GRMZM2G118800, Figure 4B); and previous studies found that ALDH activity was required for male fertility in plants. We also found two GDSL lipases in PPI network, coincidentally, MS30 was reported to encode a GDSL lipase. Thus, these KAS II, ALDH and GDSL lipases genes would be could be helpful to identify the causal gene.

We have added this information in the revision. See details in Page 5, lines 152-162, Page 7, lines 218-221, Page 8, lines 243-258.

Point 2: P7, R225: “…For instance, aldehyde dehydrogenase showed a remarkably down regulation in dyads (70-fold) and tetrad (61-fold) stage of ms305 anther…” This data is not presented in the manuscript. Table S2 shows only the DEGs that are common for the three studied anther development stages. The authors should provide additional supplemental tables with the identified DEGs in each developmental stage. In addition, both RNA-seq and lipidomics data should be submitted to public databases and accession numbers cited in the manuscript.

Response 2:

We have revised this sentence as “Notably, an aldehyde dehydrogenase gene (ALDH, GRMZM2G090245) in common DEGS showed the highest down regulation in dyads (log FC 8.37) and tetrad (log FC 7.84) stage of ms305 anther (Table S2)”, the “ratio” was corrected as “Fold change (FC)” in Table S2. See details in Page 4, line 138-140, and Page 12, Table S2.   The identified DEGs in cuticle and lipid metabolism pathways were provided in Table S3.

RNA-seq raw data generated in this study are deposited at NCBI SRA database (https://dataview.ncbi.nlm.nih.gov/object/PRJNA613313?reviewer=cfcpa0uq9rgbq0n60t1uouj7h6)

 Lipidomics data were provided in Table S7.

Point 3: P6, R205: “…However, the abortion characteristics of ms305 were different from Ms33,…” Please, describe the differences in more details. The observed phenotype of ms305 should also be compared to other known MS mutants with affected lipids biosynthesis and the differences highlighted and discussed to further support the novelty of ms305 mutant.

Response 3:

We have revised this paragraph as “The abortion characteristics of Ms33, which showed no difference in anatomical structure or meiotic events in the fertile and sterile plants before the tetrad stage, as well as the severe degradation of the tapetum in sterile plants at the early uninucleate stage, were completely opposite to those of ms305 that delay the degradation of tapetum at this stage.’ See details in Page 7, lines 224-228.

The abortion characteristics of MS30, MS33, IPE1 and APV1 in maize, whose mutants affected lipids biosynthesis, were described, such as the swollen microspore and the decreased vacuole of APV1 at the vacuolated stage, severe degradation of the tapetum in IPE1 and an obvious abortion of MS30 first appeared after vacuolated stage, these were, however, different from ms305 that the development of four somatic layers (epidermis, chamber wall, middle layer, and tapetum) were delayed in microspore stage.  See details in Page 8, lines 243-246.

Point 4: English style and grammar needs further improvement. See some more specific comments below:

-P2, R66: regard -> regarded

-P2, R67: “In spite lipid metabolism and the enzymes involved in those pathways are important for the fertility of plant male gametes, the mechanism underlying the lipid metabolism regulating anther fertility in maize remains unclear.”- Not clear, consider revising this sentence

-P2,R73: delete “genic”

-P5, R144: “which confirming”->”which confirms”

-P6, R199: “…So far, no literature reported on maize abortion characteristics is consistent with that of male sterile mutants ms305.”-Consider revising to clarify

-P6, R205 “…which showed a severe degradation of the tapetum at the early uninucleate stage [32]…”-Check the reference [32}. Seems to be incorrect in the context.

-P7, R237: hypothesis-> hypothesize

-P8, R265: “…sterile blade was used to quickly break down the spikes in the soft-smelling place.” Please clarify what is soft-smelling place.

-P8, R268: was->were

Response 4:

P2, R66: “regard” was corrected as “regarded”

P2, R67: “In spite lipid metabolism and the enzymes involved in those pathways are important for the fertility of plant male gametes, the mechanism underlying the lipid metabolism regulating anther fertility in maize remains unclear.”

This sentence has been revised as “In spite lipid metabolism pathway and its associated enzymes are important for the fertility of plant male gametes, however, the mechanism underlying the lipid metabolism regulating anther fertility in maize remains unclear”. See details in Page 2, lines 66-68.

P2, R73: “genic” has been deleted.

P5, R150: “which confirming” has been corrected as “which confirms”.

P6, R199: “…So far, no literature reported on maize abortion characteristics is consistent with that of male sterile mutants ms305.”

This sentence has been revised as “To the best of our knowledge, no mutant with the same abortion characteristics as ms305 has been reported in maize.” See details in Page 7, lines 215-216.

P6, R205 “…which showed a severe degradation of the tapetum at the early uninucleate stage [32]…”-Check the reference [32}. Seems to be incorrect in the context.

The abortion characteristics of Ms33 was reported by Zhang, L., et al., which should be the reference [39] in this study. We made a mistake in this citation, and is has been corrected, see details in page 7, line 227.

P7, R235, R268: “hypothesis” has been corrected as “hypothesize”.

P9, R265: “soft-smelling place” was revised as “soft feeling place”, we think this is more accurate. See details in page 9, lines 294-295.

P8, R268: “was” was corrected as “were”.

Round 2

Reviewer 2 Report

My previous concerns have been addressed by the authors. No comments on this version.